# Meat Substitute Development from Fungal Protein (*Aspergillus oryzae*)

**DOI:** 10.3390/foods11192940

**Published:** 2022-09-20

**Authors:** Olasky Gamarra-Castillo, Nicolás Echeverry-Montaña, Angelis Marbello-Santrich, María Hernández-Carrión, Silvia Restrepo

**Affiliations:** 1Department of Chemical & Food Engineering, Universidad de Los Andes, Bogotá 111711, Colombia; 2Laboratory of Mycology and Phytopathology, Universidad de Los Andes, Bogotá 111711, Colombia

**Keywords:** burger patty, food formulation, electronic tongue, fungal meat analog

## Abstract

The aim of this research is to develop burger patties from fungal protein. For this purpose, to maximize fungal biomass production, an optimization of the growth medium was initially carried out by testing different carbon sources and its proportion with nitrogen. Subsequently, for the design of the fungal patties, the effect of different flours, binders, and colorants on the properties of texture, water retention capacity, and color were tested, with a traditional animal-based burger patty as a control. Based on the first results, two optimal formulations were chosen and analyzed using an electronic tongue with the same control as reference. The conditions that maximized biomass production were 6 days of incubation and maltodextrin as a carbon source at a concentration of 90 g/L. In terms of product design, the formulation containing quinoa flour, carboxymethylcellulose, and beet extract was the most similar to the control. Finally, through shelf-life analysis, it was determined that the physical characteristics of the fungal meat substitute did not change significantly in an interval of 14 days. However, the product should be observed for a longer period. In addition, by the proximate analysis, it was concluded that fungal patties could have nutritional claims such as rich content in protein and fiber.

## 1. Introduction

Burger patties are an ultra-processed food of animal origin widely consumed because of their flavor characteristics and quick preparation, but the production of this food is highly polluting [1]. Many greenhouse gases are emitted into the atmosphere in livestock production [1,2]. About 51% of the total greenhouse gases are emitted by human activities, i.e., carbon dioxide (CO_2_), nitrous oxide (N_2_O), methane (CH_4_), and ammonia (NH_3_), causing acid rain and acidification of ecosystems [1,2]. Moreover, the international cancer agency of the World Health Organization (WHO) has classified the consumption of red meat (particularly processed meats) as carcinogenic [3].

Considering the problems described above, there is a need to decrease animal meat consumption from diets and therefore from products such as burger patties. Among possible meat analogues, mycoprotein is a protein-rich food obtained from the mycelia of filamentous fungi [4]. This alternative is generally 40–52% less polluting than livestock farming [5]. Filamentous fungi can efficiently convert carbohydrates and overall nutrients from the growth media into protein and their efficiency can vary depending on the carbon source and its concentration [4]. It has been found that mycoprotein consumption has potential satiety effects and would allow for reducing energy intake in subsequent meals [6]. It improves cholesterol and low-density lipoprotein profiles [6]. Finally, it is a useful and bioavailable source of protein that can help stimulate muscle protein synthesis [6]. An advantage of working with a filamentous fungus is that the texture may have a meat-like (fibrous) texture, making the transition to a more sustainable diet less drastic in terms of organoleptic characteristics [6,7].

On the other hand, it is important to note that other agents such as flours, binders, fats, water, and seasonings play an important role in burger product properties, such as water holding capacity, juiciness, and firmness [8,9]. Flours, depending on their carbohydrate content, have the function of improving the texture and consistency of the product. Among the most commonly used are rice (*Oryza sativa*) and quinoa (*Chenopodium quinoa*) flours [8,10]. Binders provide gelling and thickening properties and contribute to product stability; carboxymethyl cellulose and the enzyme transglutaminase stand out in this group [8,10]. Fats improve juiciness, tenderness, and palatability [8,10]. Solid fats, such as coconut oil (*Cocos nucifera*), give the appearance of natural juiciness of the meat and liquid fats, such as canola oil (*Brassica napus*), and improve palatability [8,10]. Water has a relevant role since it determines product density, participates in biochemical reactions (e.g., protein cross-linking), and acts as an energy transfer medium [8]. Finally, other ingredients enhance flavor and, considered the most relevant, the color of the product, since the consumer’s purchase intention usually depends on the latter [10]. The color of meat products changes during cooking, and therefore in analogous products, the aim is to obtain similarity of color before, during, and after cooking. For this purpose, beet extract (*Beta vulgaris*) and annatto (*Bixa orellana*) are often used [8,10].

This study aimed at producing a burger patty using the biomass produced by the filamentous fungus *Aspergillus oryzae*. This fungus is aerobic, and belongs to the genus *Aspergillus*, subgenus *Circumdati* section *Flavi* [11]. It can be isolated from soils and plants, particularly in rice, and can grow at a pH of 2–8, with 6 being its optimum pH [11]. This fungus reproduces asexually by the production of conidia [11]. The choice of this fungus was based mainly on its food safety, since it is recognized as a non-pathogenic fungus and has been approved for use in food by the Food and Agriculture Organization of the United Nations (FAO) [11,12]. To reach the main aim to produce a fungal-based burger patty, the growth conditions of *A. oryzae* were optimized by evaluating the effect of the carbon source and its ratio with nitrogen on biomass production. Following this, different formulations were tested by changing the binder, the flour, and the spice that provides color, analyzing their influence on the properties of texture, color, and water holding capacity (WHC), comparing it with burger patties of animal origin. A first screening was carried out where the two best formulations were selected and then, through a sensory analysis, the optimum formulation for this hamburger was chosen. Finally, this formulation was characterized by proximate and shelf-life analysis.

## 2. Materials and Methods

### 2.1. Microorganism Maintenance

The *Aspergillus oryzae* strain was obtained from the American Type Culture Collection (ATCC), reference ATCC 10124. It was maintained on a malt extract medium composed of malt extract 20 g/L, glucose 20 g/L, peptone 10 g/L, and agar 17 g/L. It was kept in an incubator (Binder World FD023UL) at 30 °C and subcultured every 14 days.

### 2.2. Culture Medium Optimization

The liquid medium used to analyze fungal growth was composed of citric acid 2 g/L, magnesium sulfate heptahydrate (MgSO_4_7H_2_O) 2 g/L, potassium dihydrogen phosphate (KH_2_PO_4_) 2 g/L, ammonium sulfate (NH_4_SO_4_) 3 g/L, calcium chloride (CaCl_2_) 0.8 g/L, yeast extract 5 g/L, and carbon source (maltodextrin or glucose) at different concentrations [13]. The inoculum concentration was 6·10^5^ conidia/mL in order to obtain a higher biomass with larger pellet diameter in accordance with results found in the literature for a fungus of the same genus, *Aspergillus niger* [14].

The carbon source and the ratio of carbon to nitrogen (yeast extract and ammonium sulfate) were considered in order to evaluate their influence on biomass production. As carbon sources, glucose and maltodextrin were evaluated at a carbon to nitrogen ratio of 15:1, 20:1, and 30:1. These levels were chosen to test the behavior of the microorganism with the carbon source, considering that biomass production can be inhibited by exceeding carbon source concentration [4].

Physical parameters such as stirring speed (100 rpm) and temperature (30 °C) were kept fixed and chosen according to previous studies [8,15]. Incubation time was determined by performing a growth curve for 10 days with a carbon–nitrogen ratio of 20:1 and using glucose as a carbon source.

Biomass production was determined by drying the recovered mycelium at 60 °C in a dehydrator (Deni Food Dehydrator 7100, Keystone Manufacturing, Rochester, PA, USA) until constant weight was reached. Efficiency was defined as the ratio of the amount of biomass obtained to the amount of carbon source entering the system (Equation (1)):(1)$$\varepsilon =\frac{weight_{Biomass}}{weight_{Carbon\;source}}\;$$

The determination of protein content was carried out by the Kjeldahl method proposed by The Association of Official Agricultural Chemists (AOAC) [16]. This indirect method of protein quantification consists of measuring total nitrogen multiplied by an empirical factor of 6.25 (under the assumption that proteins are composed of 16% nitrogen) [17]. Once optimal levels of biomass production were found, a new growth curve was made to observe the behavior of the fungus. On this occasion, in addition to the biomass obtained, carbon consumption was measured. The carbon concentration was measured by the Dubois method, which corresponds to a quantification of total sugars [18]. A calibration curve was performed with the carbon source that maximized biomass production as a standard. Finally, the total sugar content was measured using a spectrophotometer (Thermo Scientific Genesys 10 UV-Vis) at a wavelength of 490 nm [18].

### 2.3. Bioreactor Performance

Fermentation was carried out in a 5 L bioreactor (BioFlo 110, New Brunswick Scientific, Poway, CA, USA). The reactor inoculum was a 48-h fungal pre-culture in a 500 mL medium volume and an initial inoculum of 1.95 · 10^7^ conidia/mL The growth culture medium used had the conditions that optimized biomass production. An air flow enriched with 10% oxygen was used to maximize biomass yield at a rate of 1.5 L/min [19].

### 2.4. Safety and Quality Control

Biomass sample was extracted just after fermentation and inoculated on malt extract agar, as described above, to check the quality of the ferment. It was incubated for 7 days in an incubator (FD023UL, Binder, Tuttlingen, Germany) at 30 °C and then macroscopically compared with the *A. oryzae* culture, in order to ensure that there were no contaminations during the fermentation process. After the macroscopic verification, a plug of the fungus was taken and inoculated in a solid coconut (*Cocos nucifera*) medium composed of an equimolar mixture of water and coconut cream obtained locally (protein 2.5 g, carbohydrates 9.5 g and fats 22 g per 100 mL of the product according to the company’s report) and 15 g/L of bacteriological agar, in order to corroborate the absence of mycotoxins [20]. This was incubated for 5 days in the dark and then observed under UV light using a gel documenter (Gel Doc XR 170-8170, Bio-Rad, Hercules, CA, USA) [21].

### 2.5. Biomass Extraction and Product Formulation

After fermentation, RNA reduction and centrifugation were necessary. This is of great importance since RNA contains purines that increase the amount of uric acid in the body, causing (in large quantities) inflammatory diseases [4]. The RNA reduction process consisted of heat shocking at a temperature of 73 °C for 35 min using a shaker bath without shaking (BS-11, Kasai, Bogotá, Colombia) [22]. The biomass was then heated to 90 °C in the same equipment in order to inactivate mycelial growth and, finally, it was centrifuged (Sorvall legend XTR, ThermoFisher Scientific, Waltham, MA, USA) at 4000 rpm for 15 min at room temperature, and the precipitate was recovered and refrigerated at 4 °C for further processing [22].

The design of the product was barbecue style (BBQ) and, therefore, in the formulation approach, the seasonings that give the BBQ characteristic flavor were left fixed. Different formulations were prepared, iterating between two alternatives of flour, binder, and ingredients that provide color. The resulting factorial design was 23. For the first factor, flour, quinoa flour (*Chenopodium quinoa*), and rice flour (*Oryza sativa*) were used; for the second factor, binder, carboxymethyl cellulose, and the enzyme transglutaminase (microbial) were used; finally, for the third factor, color, beet extract (*Beta vulgaris*), and annatto (*Bixa orellana*) were tested. For the preparation of beet extract, 100 g of beet were blended in 100 mL of water. Table 1 shows the proportions of the product ingredients.

To prepare the patties, the ingredients were manually mixed by hand, ensuring that they were well combined.

### 2.6. Sample Preparation and Processing

First, 100 g of product was prepared for each formulation and divided into 10 portions of 10 g each. From there, it was divided into two parts, and 5 portions were destined for texture evaluation and the other 5 for water retention capacity and color. The tests were carried out on both raw and cooked product. For cooking, an electric stove was used at its maximum level with a non-stick pan for 10 min per side, confirming that the internal temperature of the hamburger reached 75 °C, measured with a thermometer to guarantee the safety of the product. Finally, it was allowed to rest for 15 min before analysis [3]. Animal-derived burger meat was used as a control, obtained from supermarkets. Its brand was Deli^®^ Gourmet and it was cooked following the same protocol described for the fungal-derived burgers.

Based on the results of the texture, color, and water holding capacity tests, the two formulations most similar to the control were chosen. These formulations were analyzed with the electronic tongue (TS-5000Z, Insent, Atsugi, Japan) to determine the best one with a sensory profile closest to that of the control. Finally, the selected formulation was subjected to proximate and shelf-life analysis. All aliases used are listed in Table 2.

### 2.7. Physicochemical Analyses

Product color was measured using a colorimeter (CR-20, Konica Minolta, Tokyo, Japan), calibrated with white ceramic. The color space was CIEL*a*b*, therefore, luminance (L*), a*, and b* were measured. Illuminant D65 and a 10° standard observer were used. Measurements were taken on the raw product and compared with the control; 5 replicates were made. With the data obtained, purity, hue, and overall color difference were calculated (Equations (2)–(4)).
(2)$$\mathrm{Purity}=C_{\mathrm{ab}}^{*}=\sqrt{a^{*}{}^{2}+b^{*}{}^{2}}$$
(3)$$\mathrm{Hue}=h_{\mathrm{ab}}^{*}=\arctan\left(\frac{b^{*}}{a^{*}}\right)$$
(4)$$\mathrm{Overall}\;\mathrm{color}\;\mathrm{difference}\;=\Delta E^{*}=\sqrt{{\left(\Delta L^{*}\right)}^{2}+{\left(\Delta a^{*}\right)}^{2}+{\left(\Delta b^{*}\right)}^{2}}$$

Concerning overall color difference, it was taken into account that for ∆E* < ±3, the differences are small and for ∆E* > ±5, the differences are perceptible to the human eye [23]. For water holding capacity, 10 g of the sample was weighed, cooked, and reweighed, and 3 replicates were done. Cooking loss and water holding capacity (WHC) were calculated with the collected data (Equations (5) $\%\mathrm{Cooking}\;\mathrm{loss}=\frac{{\mathrm{Weight}}_{\mathrm{raw}}-{\mathrm{Weight}}_{\mathrm{cooked}}}{{\mathrm{Weight}}_{\mathrm{raw}}}*100\%$ and (6)).
(5)$$\%\mathrm{Cooking}\;\mathrm{loss}=\frac{{\mathrm{Weight}}_{\mathrm{raw}}-{\mathrm{Weight}}_{\mathrm{cooked}}}{{\mathrm{Weight}}_{\mathrm{raw}}}*100\%$$
(6)WHC=100%−%Cooking loss

### 2.8. Texture

Hardness testing was performed with the texture analyzer (TA.HDplusC, Stable Micro Systems, Godalming, UK). The sample was molded in a cube shape of 1 cm × 1 cm × 1 cm × 1 cm and axially compressed with a 35 mm flat pressure adapter, at a spindle speed of 1 mm/min, a deformation of 75% and with a load of 5 g [3]. This procedure was performed both for the raw and cooked product.

### 2.9. Electronic Tongue

The electronic tongue ((TS-5000Z, Insent, Atsugi, Japan) was used to obtain the sensory profile. Three replicates and two repetitions were made. Sample preparation consisted of taking 10 g of the product and grinding it for 1 min using a blender. To make the resulting paste, 40 mL of water at 40 °C were added until the paste reached this temperature. Then, 50 g of the paste was weighed and 4 times its weight in water at 40 °C was added. It was mixed for 1 min, and it was confirmed that it was well-mixed. Finally, it was centrifuged at 3000 rpm for 10 min, filtered, and only the aqueous phase was recovered for analysis. The sensors used were CT0 for salinity, AAE for umami, CA0 for acidity, C00 for bitterness, AE1 for astringency, and AAE and C00 for richness.

### 2.10. Proximal Analysis

Protein quantification was performed using the Kjeldahl method proposed by the AOAC described previously [16]. For lipid content, the biomass was assumed to contain 3% lipids as reported in the literature [6]. Therefore, according to the formulation, a theoretical lipid content was found since it is highly dependent on the amount of oil added. The amount of ash was found by igniting 2 g of sample in a muffle (F62700, Barnstead Thermolyne, Dubuque, IA, USA) at a temperature of 500 °C for 4 h [3]. To determine the moisture, 5 g of sample were taken and placed in a forced convection oven at 105 °C for 16 h. The crude carbohydrate content was determined by macromolecule balance [3].

### 2.11. Shelf-Life

Shelf-life was determined by storing the product under refrigeration (4 °C) and samples were taken at 0, 7, and 14 days. Changes in physical characteristics were evaluated. Physical characteristics were tested for texture, color, and moisture as previously described.

### 2.12. Statistical Analyses

For the analyses of all data, an analysis of variance (ANOVA) was performed with a significance of 5%. The influence of each factor and interaction was analyzed for the optimization of the medium to find the optimal operating points. The Tukey test and Dunnett’s test were used to compare each formulation with each other and with the control, respectively, for the formulation of the meat type product. The Minitab^®^ statistical program (21.1.1) (Minitab HK Limited, Hong Kong, China) was used for all analyses.

## 3. Results and Discussion

### 3.1. Growth Medium Optimization

The optimal incubation time of the fungus was found to be 9 days since it begins its stationary phase after this time. The growth curve can be seen in Figure 1, which shows the growth for 10 days. It was evident from the results (Table 3) that the yield of biomass was the highest in the experiment run with the alias M (−1). The yield, in this case, refers to the conditions where the greatest amount of biomass is obtained with the least amount of carbon source.

In terms of protein content, it was observed that the experimental run with the highest yield reached 17.1%, which presents an advantage in terms of nutritional content regarding existing products from the Quorn brand (producer of mycoprotein with the *Fusarium venenatum* microorganism), which has around 12.8–14.5% protein in its products [6,22]. It is important to keep in mind that the recommended daily protein intake value for adults between the ages of 18 to 65 years is 0.8 g of protein per kg of a person’s weight [24]. Thus, considering the consumption of the biomass obtained by the M (−1) run, a 70 kg adult person should consume at least 327 g per day of this mycoprotein to achieve the recommended protein intake.

Statistical results analysis showed that the interaction of factors significantly affected the biomass production. When using glucose as a carbon source, as the proportion of glucose increases, the biomass recovered increased. On the other hand, using maltodextrin in low amounts resulted in a greater amount of biomass than using high proportions. It is therefore concluded that biomass production was maximized with the use of maltodextrin at a ratio of 15:1. If glucose is to be used, a high proportion of glucose should be used to maximize biomass.

A standard curve for total sugars was performed to determine carbon consumption during fungal growth. The linear regression of the standard curve follows the equation $\left[A\right]=0.0015*\left[\frac{mg}{L}\right]+0.0686$ with an R^2^ of 0.9865. When analyzing the carbon consumption and the growth of the fungus, it is highlighted that the microorganism reached its stationary phase from the sixth day (Figure 2). At the same time, it is observed that sugar consumption was not complete; in this sense, the initial amount of maltodextrin could be reduced to 90 g/L and the incubation time to six days. Regarding the incubation time, it was consistent with previous reports for the growth of *A. oryzae,* which is usually in the range of four to six days [25,26,27].

### 3.2. Quality Control

The biomass batches used in the experiment were grown in malt extract agar prior to their use in the formulation and were checked for contamination by comparing their macroscopic appearance with cultures of *A. oryzae*. Those batches that matched in macroscopic appearance were used for the elaboration of the product. It was found that the *A. oryzae* strain used does not fluoresce under UV light, indicating that the strain is not a toxin producer. This result is in agreement with the literature, since this fungus is recognized as non-pathogenic and has been approved for use in food by the Food and Agriculture Organization of the United Nations (FAO, Rome, Italy) [8,9].

### 3.3. Product Formulation

The different formulations of the meat-type products were prepared according to the experimental design. These formulations were tested for color, texture, water holding capacity, electronic tongue, and shelf-life analysis, as indicated in the methodology. In the Appendix A, the appearance of the different formulations of the raw and cooked product can be observed.

### 3.4. Color

Lightness (L*), redness (a*), and yellowness (b*) parameters were generally affected by the colorant and type of flour used. Briefly, it was found that when beet extract was used as the coloring agent, the a* and b* values were closest to the control. Although no result for a* was significantly like the control, those integrating beet extract were the closest. On the other hand, the results for b* were significantly similar to the control for some of those preparations that included beet extract in their formulation. It is important to highlight that each flour had a base color, the rice flour was white facilitating color fixation, whereas the quinoa flour was darker and yellowish. In terms of lightness (L*), it is important to consider that, according to the literature, this depends to a large extent on the amount of water and fat contained in the formulation [28]. This was evidenced by the fact that, except for QCR, all the formulations that included beet extract had higher brightness than those that included annatto; this is because the beet extract contained water in addition to the pigment, which increased the humidity of the product and therefore its brightness. In particular, it was found that the QTA, ACA, and ATR formulations were significantly similar to the control with respect to brightness. The results for L*, a*, b*, hue (h_ab_*), purity (C_ab_*), and color difference (∆E) are given in Table 4.

The hue values (h*_ab_) did not exceed 90°, indicating that the formulations presented a reddish hue that is characteristic of animal origin meat. However, the tone results were not significantly equal to the control meat. In terms of color purity (C*_ab_), it was evident that the formulations containing annatto were characterized by higher purity, which indicates that color saturation was higher. This is not desirable since having a very intense reddish color could cause a sensation in the consumer that the product is artificial and, therefore, unnatural. On the part of the beet extract, it was possible to obtain a reddish color and a lower color intensity, close to the control, thus generating an appearance closer to what is conceived in traditional meat or animal origin meat.

Obtaining a color close to traditional meat is important to impress consumers and develop expectations before consumption. In this context, the general appearance of meat substitutes should resemble known meat products in order to create positive expectations [29]. In general, the overall difference in color between the hamburger formulations regarding to the control was lower when beet extract was used as a colorant. It was therefore concluded that by using beet extract it is possible to obtain a similar color to traditional meat or animal-derived meat.

### 3.5. Water Holding Capacity

The water holding capacity of all formulations was generally found to be high (Table 5), which is beneficial since it suggests that the texture is smooth and the mouthfeel is juicy; desired characteristics in this type of product [8]. The factors significantly affecting this property were found to be the type of flour and the binder. Regarding the binder, it can be concluded that when using CMC, it was possible to obtain a higher water holding capacity, similar to the control. Regarding the flour, it was found that quinoa flour contributed to this property since, when using it, the water retention capacity increased with respect to when rice flour was used. This may be due to the fact that quinoa has an important protein content that contributes to texture and dough formation [12,30].

### 3.6. Texture

The hardness responses obtained in the formulations were high and very close to the control, as shown in Table 5. This is important since studies on consumer preferences have shown that consumers are more willing to consume meat analogs when they imitate the texture of animal origin meat [8]. Therefore, obtaining a texture similar to the control is of utmost importance since this aspect has been considered the most relevant and challenging when developing meat substitutes [29].

According to the statistical analysis, it was found that this property was affected by the type of binder and colorant, together with the interaction between both factors when raw. On the other hand, when cooked, only the interaction of the three factors (flour, colorant, and binder) was significant. CMC favored the hardness and improved the water retention capacity of the product, which agrees with other authors [31]. Regarding the colorant, it is important to keep in mind that beet extract not only adds pigmentation to the patty, as in the case of annatto, but also water. This indicates that the addition of water plays an important role in the texture of the product as it decreases the hardness of the product [31].

### 3.7. First Screening

Based on the information gathered on texture and water holding capacity, it was identified that CMC was the binder that offered the best texture and water holding capacity in the formulations. In addition, with the results obtained for color, it was found that when beet extract was used as a colorant, the final product presented a color very similar to the control. For this reason, it is concluded that for the binder and color factors, the optimum levels were carboxymethyl cellulose and beet extract.

For the choice of the optimum flour, it was decided to conduct an analysis in the electronic tongue to determine which flour achieves a sensory profile closer to the control; for this point, the use of carboxymethyl cellulose and beet extract was fixed.

It is important to highlight that the use of beet extract has been chosen in other analogous meat brands such as Beyond Meat™, because in addition to providing a similar tonality to raw animal meat, it is not as stable to heat, which is also the case with animal meat [8]. It has been observed in other studies that beet extract under heat loses red pigmentation and increases yellow pigmentation [32], thus generating brownish tones that can also be found in animal meat when cooked.

Regarding CMC, its function as a binder and thickener as reported by other authors [31,33,34] was confirmed by the results, since using it in the formulation increased water holding capacity and texture. It is important to note that this binder is also generally recognized as safe (GRAS) [34].

### 3.8. Electronic Tongue

It was found with the electronic tongue that the formulation that integrated quinoa was equally acidic, and umami had the same richness as the control meat. It differed from the control meat in bitterness, saltiness, and astringency since the burger patty developed in this research had higher values. These contrasts could be the result of the amount of seasoning added, indicating that the proportion of additional ingredients in the formulation can be reduced. The results are shown in Figure 3. Due to the great similarity between the QCR formulation and the control, it is concluded that the QCR formulation was the preferred formulation.

### 3.9. Proximal Analysis

Based on the results of the proximate analysis, the product was found to contain a high-water content and the proportion of protein provided by the product was greater than 20% of the reference value of nutrients, so that, according to resolution 810 of 2021 from Colombia, this product could obtain the declaration of excellent or rich in protein [35]. Table 6 shows the results of the proximate analysis for the selected formulation.

Although dietary fiber was not found experimentally, according to the literature, a fiber content by mycoprotein of 10% could be expected [6]. In addition, quinoa flour also has a percentage of 10% fiber content [36]. Based on the above, a theoretical fiber content can be established that would correspond to 7.5%, which would be an excellent or rich source of fiber based on the resolution 810 of 2021 from Colombia [35].

Comparing the information obtained with the nutritional table found in the control meat, it is observed that the patty developed is a higher source of protein with a low contribution of fat and total carbohydrates. However, when compared with vegan burger brands such as Beyond Meat™ or of fungal origin such as Quorn, the amount of protein in the developed burger is lower as well as the fat content. From the above, it can be concluded that, although the designed hamburger did not exceed the protein content of other vegan meats currently on the market, its protein contribution is higher than that of a traditional hamburger meat and, in addition, the amount of fat it contains is lower with respect to the control and others burger patties analyzed (Table 7). shows the nutritional information of the control, Beyond Meat™, and Quorn hamburger meat.

### 3.10. Shelf-Life Analysis

No significant changes were found in the parameters of redness (a*) and yellowness (b*) in the 14 days at 4 °C, during which time the product was analyzed. Brightness was found to increase significantly with respect to the beginning, as a result of excess water on the surface of the product due to moisture loss. Hardness decreased significantly over time between day 0 and day 14, but the changes from day 7 to day 14 were not significantly different. Although there were significant changes during the observed time, it is noteworthy that these were minimal at a global level. This indicates that the product developed is stable and its shelf-life can be prolonged. In order to mitigate the changes in hardness and water loss, it would be advisable to use mechanical agitators to promote more efficient mixing of the products and/or increase the concentration of CMC in the formulation because according to the literature, increasing its proportion also increases the hardness of the product [3]. Table 8 shows the results obtained during the observation time.

On the other hand, macroscopically, there was no presence of microorganisms as no signs of microbial growth could be detected. This result indicates that the pretreatment process of the biomass fulfilled its objective and furthermore, during the manufacturing process of the product, there was no cross-contamination that could affect the quality of the product. Figure 4 shows the image of the product on days 7 and 14 at 4 °C.

## 4. Conclusions

A burger patty with potential to have nutritional claims such as rich in protein and fiber was successfully achieved. In general, it has a higher protein content than a traditional hamburger product and its nutrient content stands out among its competitors.

It was demonstrated that the use of maltodextrin for the growth of *A. oryzae* maximized biomass production, at a concentration of 90 g/L, and an incubation time of 6 days. Regarding the design of the product, it was found that the use of beet extract achieved a similar color to meat of animal origin. In addition, carboxymethyl cellulose and quinoa flour favored the water retention capacity and texture. Furthermore, it was discovered that, under optimal levels, a product can be obtained that is not only physically like animal meat but can also have a similar sensory profile, so that the transition to a more sustainable diet does not imply sacrificing the satisfaction produced by traditional food.

For future research, it would be advisable to deepen the shelf-life analysis by increasing the observation time and, finally, to carry out a sensory analysis with potential consumers in order to evaluate the acceptability of the product in the market.

## Figures and Tables

**Figure 1 foods-11-02940-f001:**
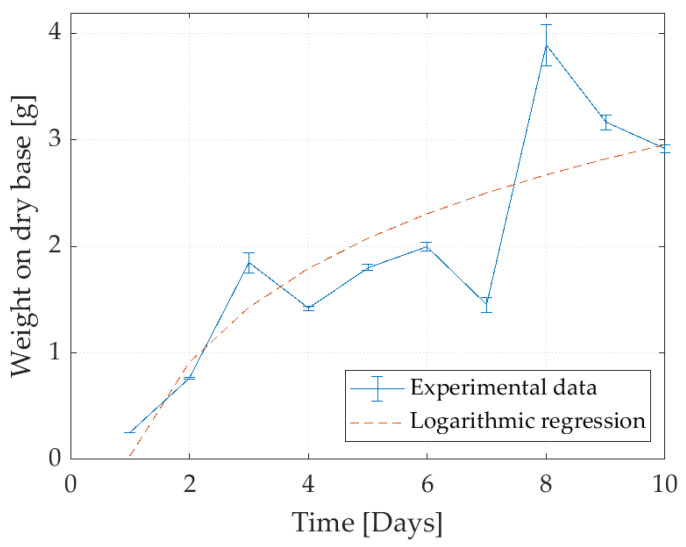
Growth curve for *A. oryzae*.

**Figure 2 foods-11-02940-f002:**
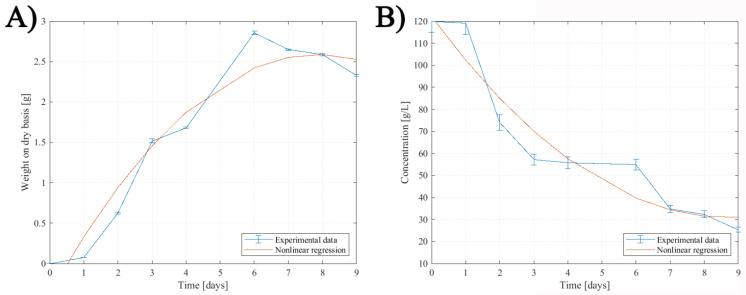
Fungus growth curve (**A**) and its carbon consumption (**B**).

**Figure 3 foods-11-02940-f003:**
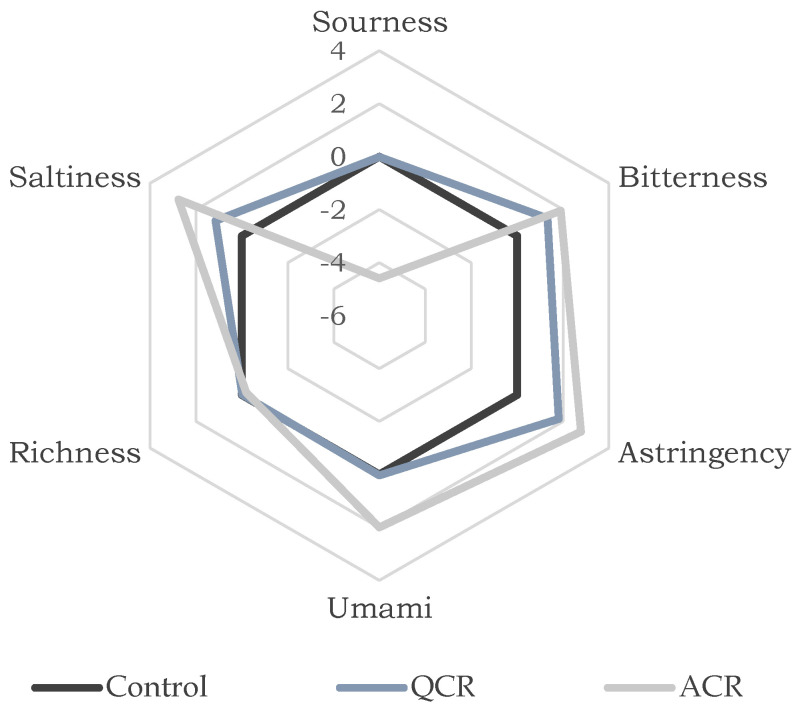
Sensory Profile.

**Figure 4 foods-11-02940-f004:**
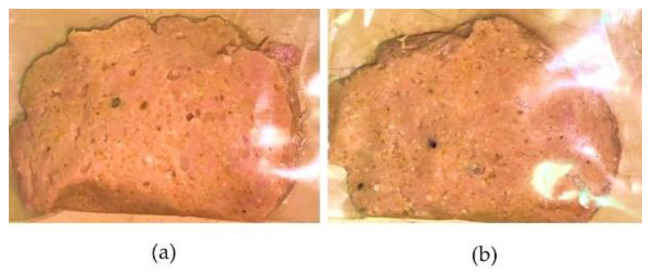
Physical appearance of the developed burger patty at 7 (**a**) and 14 (**b**) days after preparation at 4 °C.

**Table 1 foods-11-02940-t001:** General formulation composition.

Ingredients	Products Brand Name	Amount (% *w*/*w*)
Mycoprotein	Not required	55.00%
Flour	El Molino Verde	20.00%
Color	Comarrico	5.00%
Coconut Oil	El Molino Verde	3.50%
Canola Oil	Gourmet	3.50%
Binder	CMC *: Orquidea	3.00%
	TG *: WamLogic	
Mustard	La Constancia	1.93%
Worcester Sauce	Lea & Perrin	1.82%
Onion (powder)	El Rey	1.09%
Garlic (powder)	Comarrico	1.09%
Panela (powder)	El Molino Verde	0.95%
Tomato (powder)	WamLogic	0.73%
Pepper (powder)	Comarrico	0.61%
Paprika (powder)	Comarrico	0.48%
Salt (powder)	Refisal	0.36%
Vinegar	Maxima	0.36%
Soy sauce	La Constancia	0.32%
Cumin (powder)	El Rey	0.16%
Citric acid (powder)	El Molino Verde	0.05%
Ascorbic acid (powder)	El Molino Verde	0.05%

* CMC: carboxymethyl cellulose, TG: enzyme transglutaminase.

**Table 2 foods-11-02940-t002:** Alias used in experimental design.

Alias	Meaning
CMC	Carboxymethyl cellulose
TG	Enzyme transglutaminase
ACA	Formulation composed by rice flour, CMC, and annatto
ACR	Formulation composed by rice flour, CMC, and beet extract
ATA	Formulation composed by rice flour, TG, and annatto
ATR	Formulation composed by rice flour, TG, and beet extract
QCA	Formulation composed by quinoa flour, CMC, and annatto
QCR	Formulation composed by quinoa flour, CMC, and beet extract
QTA	Formulation composed by quinoa flour, TG, and annatto
QTR	Formulation composed by quinoa flour, TG, and beet extract
G (−1)	Medium composed by glucose with a C:N ratio of 15:1
G (0)	Medium composed by glucose with a C:N ratio of 20:1
G (1)	Medium composed by glucose with a C:N ratio of 30:1
M (−1)	Medium composed by maltodextrin with a C:N ratio of 15:1
M (0)	Medium composed by maltodextrin with a C:N ratio of 20:1
M (1)	Medium composed of maltodextrin with a C:N ratio of 30:1

**Table 3 foods-11-02940-t003:** Biomass and protein production.

Alias	Biomass [g]	Yield [g/g]	Protein [%]
G (−1)	2.9824 ± 0.393	4.97 ± 0.33%	30.5 ± 0.33%
G (0)	5.3319 ± 1.704	6.66 ± 1.42%	23.8 ± 1.76%
G (1)	6.5986 ± 0.042	5.50 ± 0.04%	17.3 ± 1.12%
M (−1)	8.0726 ± 1.293	13.45 ± 1.08%	17.1 ± 0.55%
M (0)	3.3466 ± 0.690	4.18 ± 0.58%	20.4 ± 0.21%
M (1)	5.9479 ± 0.107	4.96 ± 0.09%	11.8 ± 0.71%

The error reported corresponds to the deviation of the data.

**Table 4 foods-11-02940-t004:** Color parameters for raw formulations.

Alias	L*	a*	b*	$h_{ab}^{*}\;({}^{\circ})$	$C_{ab}^{*}$	ΔE
QCA	44.95 ^b^ ± 0.071	23.10 ^c^ ± 0.283	25.25 ^a^ ± 0.636	47.54 ^a^ ± 1.069	34.23 ^b^ ± 0.279	17.61 ^b^ ± 0.246
QCR	41.5 ^c,d^ ± 0.424	13.95 ^e^ ± 0.354	12.2 ^b^ ± 0.283	41.17 ^d^ ± 1.378	18.53 ^e^ * ± 0.080	4.92 ^e^ ± 0.051
QTA	42.5 ^c^ * ± 0.424	24.80 ^b^ ± 0.566	25.75 ^a^ ± 0.354	46.08 ^a,b^ ± 0.260	35.75 ^b^ ± 0.647	19.18 ^b^ ± 0.659
QTR	45.2 ^b^ ± 0.566	15.00 ^e^ ± 0.283	12.95 ^b^ * ± 0.495	40.80 ^d^ ± 0.549	19.82 ^d,e^ ± 0.538	5.69 ^e^ ± 0.342
ACA	42.8 ^c^ * ± 0.283	28.55 ^a^ ± 0.495	26.45 ^a^ ± 0.778	42.81 ^c,d^ ± 0.345	38.92 ^a^ ± 0.892	22.61 ^a^ ± 0.865
ACR	48.85 ^a^ ± 0.212	18.35 ^d^ ± 0.212	12.75 ^b^ * ± 0.212	34.79 ^e^ ± 0.136	22.34 ^c^ ± 0.295	10.28 ^c^ ± 0.079
ATA	40.95 ^d^ ± 0.212	27.65 ^a^ ± 0.354	26.95 ^a^ ± 0.495	44.26 ^b,c^ ± 0.160	38.61 ^a^ ± 0.599	22.28 ^a^ ± 0.594
ATR	42.85 ^c^ * ± 0.354	17.10 ^d^ ± 0.141	12.05 ^b^ ± 0.212	35.17 ^e^ ± 0.698	20.92 ^c,d^ ± 0.007	7.60 ^d^ ± 0.095
Control	43.35 ± 0.212	9.75 ± 0.071	13.9 ± 0.141	54.95 ± 0.469	16.98 ± 0.075	0.00 ± 0.000

^a–e^ Different letters within the same column indicate significant differences between samples according to Tukey’s test (*p* < 0.05). * Indicates non-significant differences with respect to the control according to Dunnett’s test (*p* < 0.05).

**Table 5 foods-11-02940-t005:** Results for water holding capacity for raw formulations and hardness for raw and cooked formulations.

Alias	WHC	Hardness [N]	
		Raw	Cooked
QCA	92.24 ^a^ * ± 0.37%	65.92 ^a,b^ ± 0.10	66.58 ^a,b^ * ± 0.01
QCR	91.87 ^a^ * ± 0.18%	65.64 ^c^ ± 0.06	66.47 ^a,b^ ± 0.06
QTA	88.49 ^a,b^ * ± 0.70%	65.66 ^c^ ± 0.04	66.37 ^b^ ± 0.01
QTR	88.69 ^a,b^ * ± 0.27%	65.63 ^c^ ± 0.01	66.62 ^a^ * ± 0.03
ACA	90.19 ^a,b^ * ± 1.15%	65.95 ^a^ ± 0.04	66.51 ^a,b^ * ± 0.07
ACR	90.07 ^a,b^ * ± 0.04%	65.68 ^b,c^ ± 0.08	66.56 ^a,b^ * ± 0.05
ATA	85.16 ^b,c^ ± 2.35%	65.49 ^c^ ± 0.09	66.48 ^a,b^ ± 0.11
ATR	83.70 ^c^ ± 1.13%	65.69 ^b,c^ ± 0.01	66.42 ^a,b^ ± 0.04
Control	93.33 ± 0.15%	66.61 ± 0.07	66.68 ± 0.01

^a–c^ Different letters within the same column indicate significant differences between samples according to Tukey’s test (*p* < 0.05). * Indicates non-significant differences with respect to the control according to Dunnett’s test (*p* < 0.05).

**Table 6 foods-11-02940-t006:** Proximal analysis of the fungal burger patty.

Content	Proportion (%p/p)	Portion of 100g (g)	Reference Nutrient Values for a 2000 kJ Diet (%)
Moisture	57.38 ± 2.166	57.38	--
Protein	13.13 ± 0.625	13.13	26
Fat	10	10.00	15
Ash	8.85 ± 0.634	8.85	--
Total Carb.	10.64 ± 2.992	10.64	4
Energy	--	185.05	9

-- indicates there is no related information.

**Table 7 foods-11-02940-t007:** Nutritional content of other burger patties [37,38].

Nutrients	Control	Beyond Meat™	Quorn
Energy, kcal (100 g)	200	230	244
Protein, %	12	17.7	20.5
Fat, %	11	15.93	13.3
Total Carb, %	15	4.42	8.9
Fiber, %	1	1.8	3.2
Sodium, %	2.030	0.345	1.2

**Table 8 foods-11-02940-t008:** Shelf-life analysis.

**Properties**	**Day 0**	**Day 7**	**Day 14**
Hardness	65.48 ^a^ ± 0.59	63.34 ^a,b^ ± 0.94	62.01 ^b^ ± 0.06
Humidity	56.17 ^a^ ± 0.77%	48.30 ^b^ ± 1.90%	47.37 ^b^ ± 1.60%
L*	43.00 ^b^ ± 1.41	50.70 ^a,b^ ± 4.81	55.65 ^a^ ± 0.64
a*	10.35 ^a^ ± 0.07	9.50 ^a^ ± 0.57	10.75 ^a^ ± 0.07
b*	12.75 ^a^ ± 0.21	12.00 ^a^ ± 0.99	14.10 ^a^ ± 0.42

^a,b^ Different letters within the same row mean significant differences according to Tukey’s test (*p* < 0.05).

## Data Availability

Data is contained within the article or Appendix A.

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
