# Peer review of "Meat Substitute Development from Fungal Protein (Aspergillus oryzae)"

_foods, 2022, doi:10.3390/foods11192940_

Round 1

Reviewer 1 Report

The manuscript is well written and has some significant findings. The language is clear and easy to understand. The research area and experimental design are novel and have industrial applications. The hypothesis is well stated and clearly defined. I have the following comments as-

Abstract:

I think there is a need for more clarity needed on the very general statement made in lines 8-9; its overconsumption that creates issues in the case of red meat consumption. The Authors may shift that portion into the introduction and the abstract may start after line 9.  Authors may refer to the https://doi.org/10.1080/10408398.2022.2096562. May be the value of % fits for Dutch or Portugal but not for other regions.

Keywords: Appropriate

Introduction

Appropriate and sufficient background information was provided to start the work. However, I think the first three paragraphs could be better formatted and concise and more focused on the hypothesis. In the introduction, please mention clearly meat burger patties for better understanding as in the majority of Asian countries, it is mostly famous as veg products (such as India).

Other comments

       i.          In Table 1: Please change , to . in data

     ii.          The shelf life assessment is done by only observing products physical parameters.

   iii.          Line 305: please check for error

   iv.          Table 6: please delete the % from the values in first column

     v.          Line 446-449: please check how macroscopically the microbes are assessed

Thank you so much for giving opportunity to read your research work 

Author Response

IN THE BELOW PARAGRAPHS WE WILL ANSWER POINT BY POINT

I think there is a need for more clarity needed on the very general statement made in lines 8-9; its overconsumption that creates issues in the case of red meat consumption. The Authors may shift that portion into the introduction and the abstract may start after line 9.  Authors may refer to the https://doi.org/10.1080/10408398.2022.2096562. May be the value of % fits for Dutch or Portugal but not for other regions.

THANKS, WE STARTED THE ABSTRACT IN THE LINE OF THE AIM OF THE STUDY

WE DELETED THE REFERNCE TO PORTUGAL

Introduction

Appropriate and sufficient background information was provided to start the work. However, I think the first three paragraphs could be better formatted and concise and more focused on the hypothesis. In the introduction, please mention clearly meat burger patties for better understanding as in the majority of Asian countries, it is mostly famous as veg products (such as India).

WE CHANGED SOME SENTENCES OF THE FIRST THREE PARAGRAPHS TO MAKE THE INTRODUCTION CLEARER AND WE ADDED MEAT OR ANIMAL-BASED TO THE BURGERS.

Other comments

  1. In Table 1: Please change , to . in data

DONE

  1. The shelf life assessment is done by only observing products physical parameters.

AS MENTIONED IN THE MANUSCRIPT: Changes in physical characteristics were evaluated. Physical characteristics were tested for texture, color and moisture as previously described: TEXTURE IN SECTION 2.8, MOISTURE IN SECTION 2.10 AND COLOR IN SECTION 2.7

   iii.          Line 305: please check for error

THE ERROR WAS CORRECTED

  1. Table 6: please delete the % from the values in first column

DONE

  1. Line 446-449: please check how macroscopically the microbes are assessed

WE ADDED THE SENTENCE: On the other hand, macroscopically there was no presence of microorganisms as no signs of microbial growth could be detected.  INDEED, WE ONLY INSPECTED THE PRODUCT BY VISUAL OBSERVATION

Reviewer 2 Report

I reviewed the manuscript entitled, Meat substitute development from fungal protein (Aspergillus oryzae). The manuscript has novelty and contributes to the field. The scientific quality and approach are appropriate

Line 17: traditional burger patty. Is it plant or animal based?

Are author compare plant-based burger with animal-based burger? What is the difference in nutritional composition of both? According to mx research experience, there are some differences in nutritional attributes of plant-based burger over animal-based burger.  Authors should address this point

Research objectives should be revised

Lines 121 to 122: eq is repeated.

Also, check all other eq. for example, section 2.7

Figure 2: quality must be improved

Table 4. Color test results for raw formulations; should be re-written as color parameters for raw formulations

Table 5. Results for water holding capacity and hardness: which results? Re-write it

Section 3.10: what is the temp for conducting shelf-life studies?

References are not according to the journal format. Please revise it.

Author Response

IN THE PARAGRAPHS BELOW WE RESPOND TO EACH COMMENT

Line 17: traditional burger patty. Is it plant or animal based?

WE ADDED ANIMAL-BASED

Are author compare plant-based burger with animal-based burger? What is the difference in nutritional composition of both? According to mx research experience, there are some differences in nutritional attributes of plant-based burger over animal-based burger.  Authors should address this point

Research objectives should be revised

WE REVISED AND CHANGED THE LAST PARAGRAPH OF THE INTRODUCTION TO EXPLAIN OR MAKE CLEAR THAT WE HAVE A MAIN AIM THAT IS TO PRODUCE A FUNGAL-BASED BURGER PATTY. THEN, TO REACH THIS OBJECTIVE WE HAD TO OPTIMIZE THE PRODUCTION PROCESSES

Lines 121 to 122: eq is repeated.

CORRECTED

Also, check all other eq. for example, section 2.7

CORRECTED

Figure 2: quality must be improved

DONE

Table 4. Color test results for raw formulations; should be re-written as color parameters for raw formulations

RE-WRITTEN

Table 5. Results for water holding capacity and hardness: which results? Re-write it

RE-WRITTEN

Section 3.10: what is the temp for conducting shelf-life studies?

WE ADDED THE TEMPERATURE, 4ºC

References are not according to the journal format. Please revise it.

REVISED